# Hydrodynamic dispensing and electrical manipulation of attolitre droplets

Yanzhen Zhang[1], Benliang Zhu[2], Yonghong Liu[3] & Gunther Wittstock[1]

Dispensing and manipulation of small droplets is important in bioassays, chemical analysis and patterning of functional inks. So far, dispensing of small droplets has been achieved by squeezing the liquid out of a small orifice similar in size to the droplets. Here we report that instead of squeezing the liquid out, small droplets can also be dispensed advantageously from large orifices by draining the liquid out of a drop suspended from a nozzle. The droplet volume is adjustable from attolitre to microlitre. More importantly, the method can handle suspensions and liquids with viscosities as high as thousands mPa s markedly increasing the range of applicable liquids for controlled dispensing. Furthermore, the movement of the dispensed droplets is controllable by the direction and the strength of an electric field potentially allowing the use of the droplet for extracting analytes from small sample volume or placing a droplet onto a pre-patterned surface.

[1] Institute of Chemistry, Center of Interface Sciences, Faculty of Mathematics and Science, Carl von Ossietzky University of Oldenburg, D-26111 Oldenburg, Germany. [2] Guangdong Province Key Laboratory of Precision Equipment and Manufacturing, School of Mechanical and Automation Engineering, South China University of Technology, 510640 Guangzhou, China. [3] Department of Mechanical and Electronic Engineering, College of Mechanical and Electronic Engineering, China University of Petroleum, Qingdao 266580, China. Correspondence and requests for materials should be addressed to Y.Z. (email: zhangyanzhen.upc@gmail.com) or to G.W. (email: gunther.wittstock@uni-oldenburg.de).

Dispensing and manipulation of small droplets is central to inkjet printing[1,2], bioassays[3–5], drug delivery[6,7], microcapsules[8–10], microreactors[11,12] and fabrication of micromechanical[13–15], -electrical[16,17] and even -biological[18–20] devices. Over the past decades, scientific and industrial communities have aimed for dispensing smaller and smaller drops. So far, small droplets have been produced by methods based on micro-orifices (or channels) or by orifice-free methods. Examples for micro-orifice-based methods comprise the piezo-driven injection[21], and the heat bubble ejection[22] widely used in the field of inkjet printing. The T-junction[23] and flow focusing[24] methods stand for channel-based methods used in microfluidics. Despite the effectiveness of the micro-orifice (or channel)-based droplet-producing devices, their typically confined geometry poses several challenges, such as high flow resistance and propensity to clogging. For the micro-orifice (or channel)-based methods, droplets were generated by squeezing the liquid out of the orifice, and this is the main reason why the size of the droplets depends largely on the size of the orifice. The currently existing micro-orifice (or channel)-based techniques (for example, inkjet printing) still encounter problems to generate droplets that are substantially smaller than the orifice from which they are ejected[25]. The size parameter $\chi = R_d/R_o$ is the ratio of the droplet radius $R_d$ and the inner radius of the orifice $R_o$. Typically $\chi$ lies between 0.5 and 3 (ref. 25), and, consequently, ultra-small orifices would be needed for the generation of sub-micrometer-sized droplets[26]. Although the electrohydrodynamic jet (e-jet) printing[27] can produce droplets much smaller than the orifice in the micrometre scale, it still meets constrains when aiming for small values of $\chi$ when $R_d$ is in the sub-micrometre range. The coupling of drop and orifice sizes translates into challenges for fabrication of fragile micronozzles and handling them, for example, prevention of clogging when dispensing highly viscous liquids or suspensions.

Besides the orifice-based techniques, small droplets can also be produced by orifice-free techniques, such as droplets splitting based on surface-wetting properties[28,29], pyroelectrodynamic-driven techniques[30] and nanowire liquid pump techniques[31]. However, those techniques evoke a number of complex technical problems that require locally tuning of wetting properties of the substrate surface[28,29], manipulation of a hot tip or an infrared laser beam[30] or nanowires[31] on a substrate.

Here we present a novel approach to dispense ultra-small droplets with controllable size. Instead of squeezing the liquid out of the nozzles, the droplets are formed by sucking back the liquid from a sessile drop initially suspended from the nozzles. In this way the size parameter $\chi$ can be decreased to 0.025 enabling the dispensing of picolitre to microlitre volumes even from a millimetre-sized orifice that is rather immune against clogging when dispensing highly viscous liquids or suspensions. Furthermore, the use of orifices with diameters of several micrometres shows the capacity of dispensing sub-micrometre sized droplets.

## Results

**Daughter droplet pinch-off regimes.** We investigate this phenomenon with a sessile surfactant-stabilized water droplet suspended from surface-treated nozzles, which are connected to a computer-controlled syringe pump for extruding droplets or draining the liquid from the droplets back into the nozzle. Figure 1 shows that by draining the liquid with different drainage rates $Q$, the droplet size $V_d$ can be controlled and pinch-off regimes can be switched. Figure 1a and Supplementary Movie 1 show that the whole pinch-off process can be divided into two stages: first, the sphere-shaped sessile drop evolves into a cylindrical column with the height being much larger than its radius due to the drainage of the liquid. Second, the liquid column breaks up due to Rayleigh–Plateau instability after the stop of the drainage and pinches off one main daughter droplet with $R_d$ much smaller than the radius of the mother droplet $R_m$. In addition, several ultra-fine satellite droplets are formed due to break-up of the liquid neck. Image analysis showed that the main droplet contains more than 99% of the total volume of the daughter droplets (last amplified panel of Fig. 1a). The presence of satellites proves that the neck was indeed broken up due to the Rayleigh–Plateau instability. The extension factor $\varepsilon$ of the liquid column, defined as the ratio of its length to circumference, is very important for the subsequent break-up. In fact, we observed that the liquid columns always break-up if $\varepsilon > 1$. In contrast, they never break-up when $\varepsilon < 1$ consistent with the classical theory of Rayleigh–Plateau instability (Fig. 1c). This regime occurs when using small $Q$. Between pinch-off regime (Fig. 1a) and no pinch-off regime (Fig. 1c), there is a transition regime in

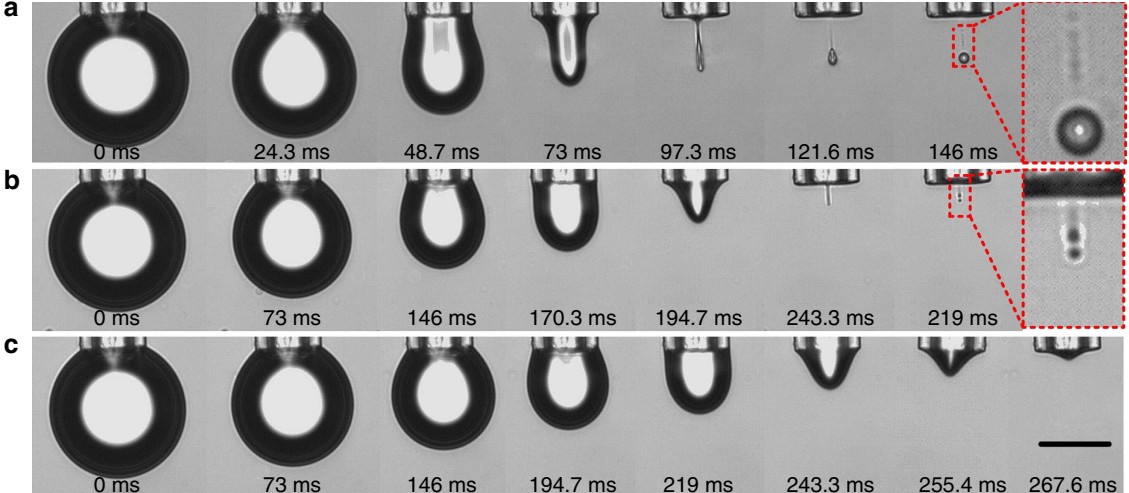

**Figure 1 | Three regimes of droplet formation.** Droplet formation at various $Q$ (**a**) 7.2 nl s$^{-1}$ (regime A, one main droplet), (**b**) 2.46 nl s$^{-1}$ (regime B, multiple droplets of similar sizes) and (**c**) 1.96 nl s$^{-1}$ (regime C, no droplet formation). The outer and inner radii of the nozzle are 23 and 15 µm, respectively, $V_m = 0.52$ nl, $\sigma_{12} = 9$ mN m$^{-1}$. Scale bar, 50 µm.

which several drops of similar sizes are formed (Fig. 1b). Supplementary Movies 2 and 3 show the dynamic process of the transition regime and no pinch-off regime.

**Dispensing ability.** Figure 2a–c shows that when dispensing water into silicon oil from orifices with $R_o$ of 550, 175 and 90 μm, the smallest droplet radii $R_d$ are about 30, 10 and 5 μm, corresponding to $\chi$ of 0.054, 0.057 and 0.055, respectively. This unique feature of this method promises the dispensing of nanometre-sized droplets from micrometre-sized orifices. Indeed, droplets with $R_d < 500$ nm are successfully generated from an orifice with $R_o = 5$ μm (Fig. 2d) approaching the limits for the optical observation (Supplementary Movie 4). So far, the smallest $\chi = 0.025$ has been obtained when dispensing glycol into silicone oil with viscosity $\mu_2 = 1,000$ mPa s (Fig. 2e). Importantly, the principle can be used to dispense liquids of higher viscosity as shown in Fig. 2f for dispensing glycerine–water mixtures into silicon oil or for dispensing highly viscous silicon oils (viscosities higher than 1,000 mPa s) in silicone elastomer (viscosity 10 Pa s). Experimental results showed that highly viscous liquids can be dispensed with similar drop sizes as water droplets. This is of great practical significance as many functional inks with high viscosity can be directly applied without pre-processing to reduce the viscosity. This can greatly simplify the use of sensitive or reactive inks for which heating as the most frequently used method for viscosity matching is often not applicable.

The small size factor $\chi$ enables generation of small droplets from large orifices. This renders the method particular suitable for dispensing suspensions with a high content of a second liquid phase or of solid particles (Fig. 2g,h). By controlling the size of the oil droplets within an oil-in-water emulsion and the size of the dispensed droplet, 'microcapsules' can be formed containing many small oil droplets (Fig. 2g). It is also possible to enclose only a limited number of oil droplets in each dispensed droplet (Fig. 2h) by either increasing the size of the oil droplets within the emulsion or by reducing $V_d$. This paves the way towards new approaches for the preparation of microcapsules with a broad range of applications[8,32,33].

**Droplet manipulation after pinch-off.** To manipulate the extremely small droplets after pinch-off, charged droplets were formed and driven in an electric field $E$ (Fig. 3a and Supplementary Movie 5). At the time of pinch-off, the daughter droplets will be charged due to the induction[34]. The charged droplets will move towards the oppositely charged substrate and attach to its surface (Supplementary Note 1 and Supplementary Fig. 1). Manoeuvring the droplets in an electric field holds great potential for precisely placing droplets on conductive or on insulating surfaces by attaching the nozzle to a three-dimensional positioning system (schematic in Fig. 3a, formed droplet array in Fig. 3b, side view of this process in Figs 2c and 3c,d). In addition to depositing small droplets on a solid surface, this method also allows manipulation of the dispensed small droplets after dispensing. First, the droplet can be taken back to the reservoir before it adheres to the substrate just by changing the polarity of the voltage between the nozzle and the substrate (Fig. 3c and Supplementary Movie 6). As shown in Fig. 3c, the initially negatively charged droplet will move upward under the action of Coulomb force after reversing the polarity of the power supply and finally merge with the mother drop. Second, by increasing the applied voltage, dispensed droplets will bounce between the mother drop and the substrate, and will horizontally follow a lateral movement of the nozzle enabling the sequential contact of a droplet with different regions of a substrate surface (Fig. 3d and Supplementary Movie 7). During the bouncing, the droplet will assume alternatingly the charge of the nozzle or of the substrate followed by a reversal of the vertical motion direction. This process is possible because there is a critical electric field strength $E_{crit}$ above which two oppositely charged drops do not coalescence[35–37].

## Discussion

During the pinch-off process, the gravity does not play a significant role since the Bond number Bo $= (\rho_1 - \rho_2)gR_m^2/\sigma_{12}$ is always smaller than 0.1. $\rho_1 - \rho_2$ is the density difference between the dispensed (1) and surrounding liquid (2), $g$ is the acceleration due to gravity and $\sigma_{12}$ is the interfacial tension between the two liquids. The deformation of the sessile drop is driven by the externally controlled drainage and the viscous force $F_v$ acting on the liquid–liquid interface. It is mainly resisted by the interfacial tension $F_\sigma$. The drainage and viscous force tend to elongate the mother drop, whereas the interfacial tension counteracts this process. This suggest the capillary number, Ca $= \gamma\mu_2 R_m/\sigma_{12}$, should have an important influence on the

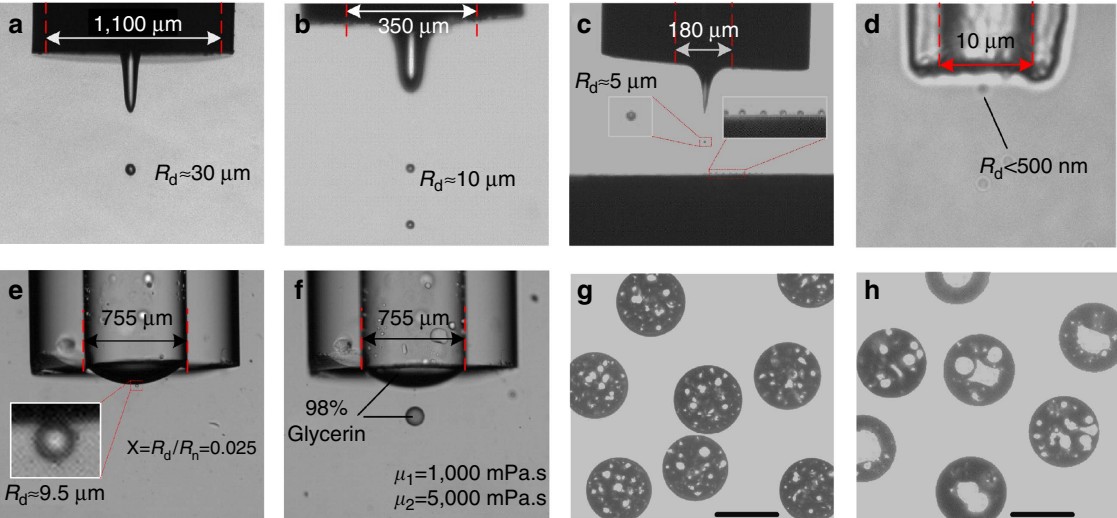

**Figure 2 | Dispensed droplets and microcapsules.** (**a–f**) Dispensing droplets from orifices with different radii shown in the panel. (**g,h**) Microcapsules obtained by dispensing silicon oil-in-water emulsion. Scale bars in **g** and **h**, 100 and 30 μm, respectively.

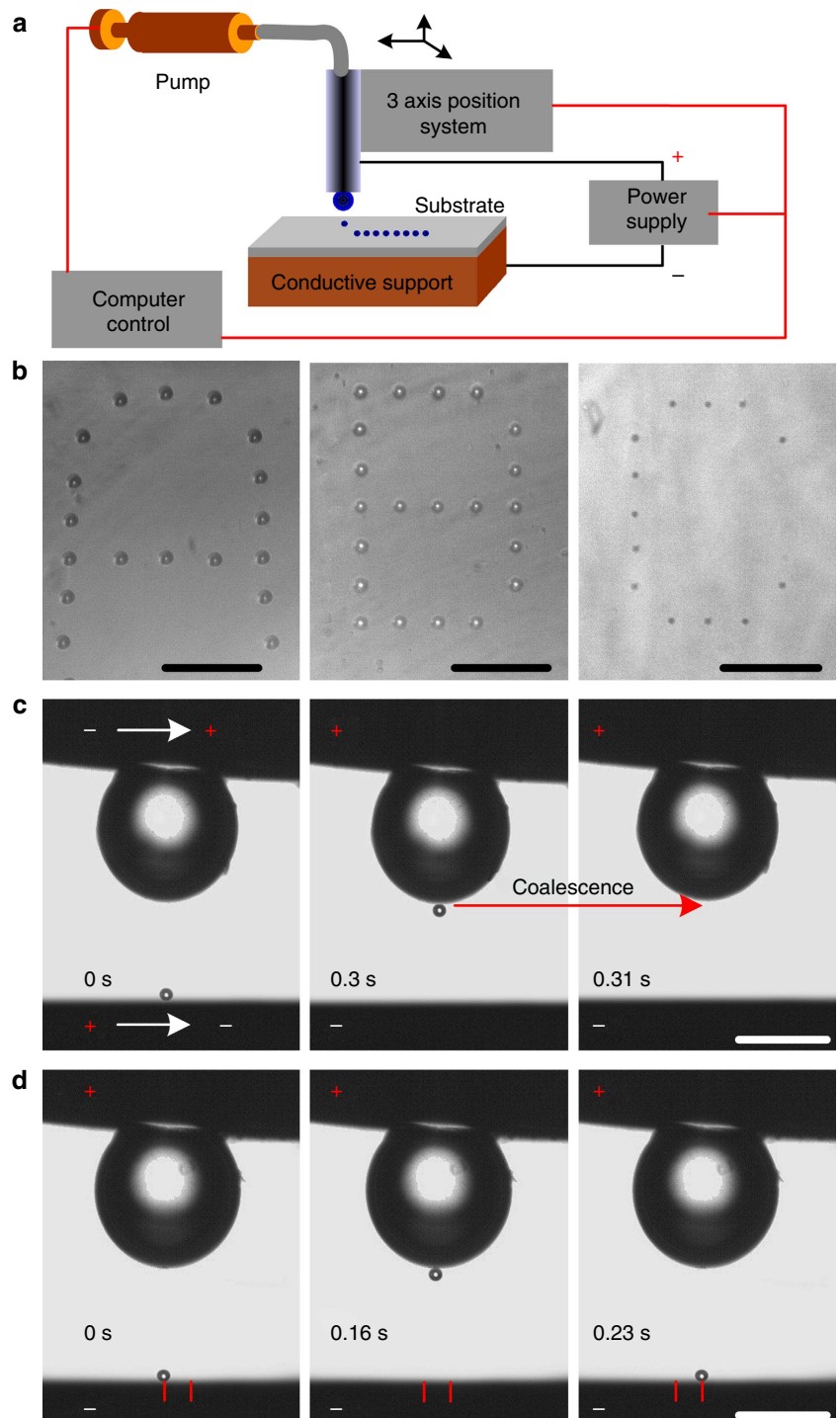

**Figure 3 | Electric field aided hydrodynamically dispensing and manipulation of small droplets.** (**a**) Schematic representation of the experimental set-up. (**b**) Characters 'written' by dispensing small droplets on the surface. The characters 'A' and 'B' were written with an orifice with diameter of 180 μm, 'C' was written with an orifice of 40 μm. The average droplet radii for characters 'A', 'B' and 'C' are about 15, 7 and 2 μm, respectively. Scale bars for 'A', 'B', and 'C', 200, 100 and 50 μm, respectively. (**c**) Merging of a dispensed droplet with the mother droplet. (**d**) Bouncing and horizontal movement of the dispensed droplets under the action of Coulomb force. Scale bars in **c** and **d**, 200 μm.

pinch-off of daughter droplets. Here $\mu_2$ is the dynamic viscosity of liquid 2 (because the viscosity of liquid 1 $\mu_1$ is much smaller than $\mu_2$ in our experiments and has very little influence on the deformation), $\gamma$ is the shear rate and can be expressed as $Q/\pi R(t)^3$, and $R(t)$ is the radius of the liquid column during the drainage process. Pursuing an analytic solution for $\gamma$ is challenging since $R(t)$ is dynamically changing during the

drainage. Nonetheless, since the daughter droplet pinch-off is much more affected by later stages of the drainage process (when $R(t) < R_o$) rather than by the earlier stages of the drainage process (when $R(t) > R_o$), we define an effective capillary number of the system $Ca = \mu_2 Q R_m / \pi \sigma_{12} R_o^3$.

The experimental results show that the pinch-off regimes, as well as the size of the daughter droplets is controlled by the

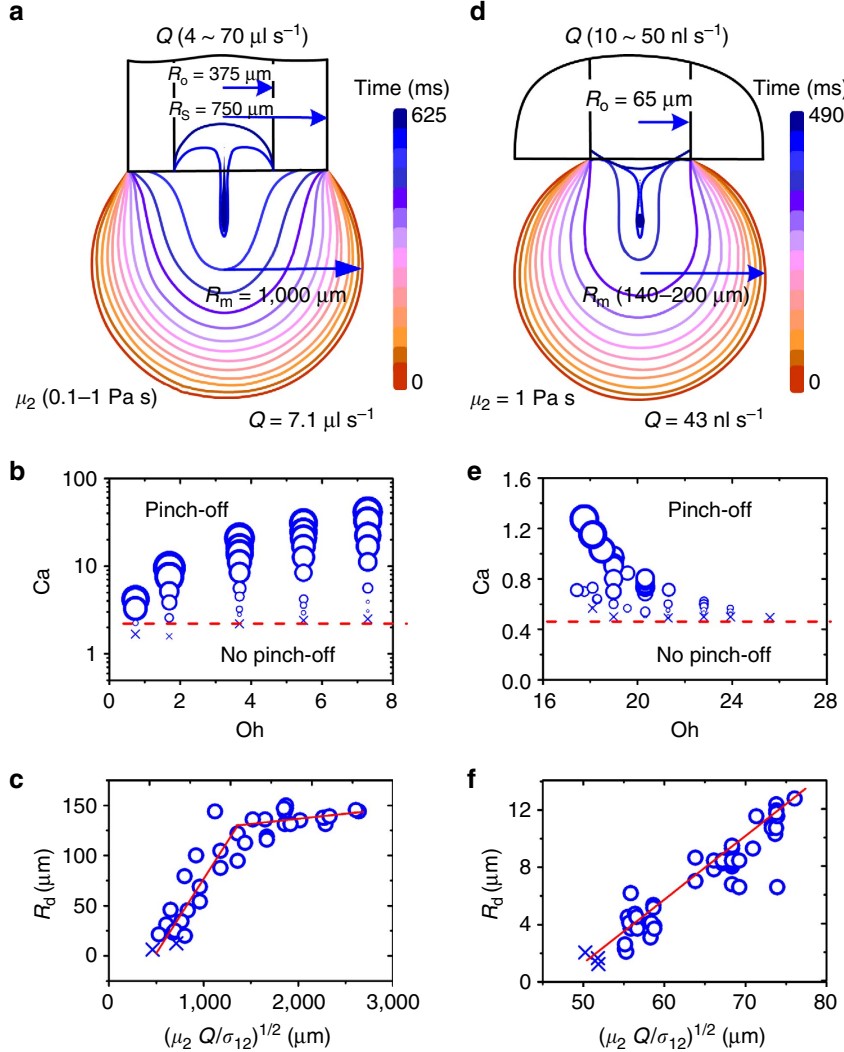

**Figure 4 | Size of daughter droplets as a function of experimental parameters. (a)** Geometry when the mother droplet is pinned at the outer radius of the capillary. The shapes were extracted from video images. **(b,c)** Experimental daughter drop sizes. **(d)** Geometry when the mother droplet is pinned at the inner radius of the capillary. **(e,f)** Experimental daughter drop sizes. The size of the symbol in **b** and **e** is proportional to the size of the daughter droplets, and the abscissa is given as dimensionless Ohnesorge number $Oh = \mu_2[(\rho_1 + \rho_2)\sigma_{12}R_m]^{-\frac{1}{2}}$ comparing viscous, inertial and interfacial forces. **(c,f)** The linear relationship between $R_d$ and $(\mu_2 Q/\sigma_{12})^{1/2}$. Crosses indicated dispensing under formation of several droplets of similar sizes (transition regime) where $R_d$ is the average radius of the several droplets formed due to the break-up of the thin liquid filaments.

capillary number of the particular system (Fig. 4). Two sets of experiments with different orifice sizes and other parameters were carried out. In the first set, microdroplets were generated in five different silicon oils, which have different viscosities (0.1, 0.25, 0.5, 0.76 and 1 Pa s), densities (varied from 1.01 to 1.09 g ml$^{-1}$) and interfacial tensions with the sessile drop (varied from 9 to 10 mN m$^{-1}$). Drainage rates varied from 4 to 70 µl s$^{-1}$. The mother drop is initially settled on the outer circle of the capillary (Fig. 4a). The three-phase contact line contracts during the liquid drainage process and finally anchored on the inner surface of the orifice after the drainage. In the second set of experiments, microdroplets were generated in one silicon oil ($\mu_2 = 1$ Pa s, $\rho_2 = 1.09$ g ml$^{-1}$, $\sigma_{12} = 10$ mN m$^{-1}$) with a stainless-steel capillary ($R_o = 65$ µm). $R_m$ varied from 140 to 200 µm and $Q$ varied from 10 to 50 nl s$^{-1}$. The mother drop is initially settled on the inner circle of the capillary and is anchored there during the entire drainage process (Fig. 4d). The evolution of the drop shapes for the two sets of experiments at drainage rates of 7.1 µl s$^{-1}$ and 43 nl s$^{-1}$ are shown in Fig. 4a,d, respectively. Within both sets, the size of the daughter droplets increases with

Ca (Fig. 4b,e). There is a critical capillary number Ca* in each geometry, which separates the regimes of no droplet formation and droplet formation (red dashed lines in Fig. 4b,e). Owing to the significant difference of the geometries, the value of Ca* for the two sets are different.

The above analysis suggests that the radius of the daughter droplets $R_d$ should be determined by the radius $R(t)$ of the liquid column just before pinch-off. During the drainage process, the liquid column is dragged by the viscous force $F_v$, which scale as $F_v \sim \mu_2 \gamma$. The shear rate $\gamma$ should scale as $Q/R(t)^3$. During the drainage $F_v$ should be balanced by interfacial tension $F_\sigma$, which equals $\sigma_{12}/R(t)$. Thus, $F_v \sim \mu_2 Q/R(t)^3 \sim \sigma_{12}/R(t)$. Consequently, $R_d$ scales approximately according to $R_d \sim (\mu_2 Q/\sigma_{12})^{1/2}$. Figure 4c,f, in which all measured $R_d$ are replotted versus $(\mu_2 Q/\sigma_{12})^{1/2}$, shows that the capillary-viscous prediction provides indeed a good estimate of the size of the daughter droplet. It should be noted that $R_d$ cannot increase unlimited with $(\mu_2 Q/\sigma_{12})^{1/2}$ (Fig. 4c). Because of $(\mu_2 Q/\sigma)^{1/2} = (R_o^3/R_m)^{1/2} Ca^{1/2}$, $R_d$ is constrained by $R_o$ and $R_m$ (Supplementary Note 2 and Supplementary Fig. 2).

This work presents a novel hydrodynamic dispensing method for small droplets and demonstrates that the pinch-off regimes, as well as the size of the dispensed droplets are determined by the capillary number of the particular system. The radius of the dispensed droplets $R_d$ can be as small as 0.025 times the radius of the orifice $R_o$. This allows dispensing nanometre-sized droplets (several hundred attolitre) from micrometre-sized orifices with high robustness. Millimetre-sized orifices facilitate dispensing nano/picolitre droplets of highly viscous liquids, emulsions or suspensions that are difficult to process by established ink-jet technologies due to the clogging of smaller nozzles. The method also exhibits remarkable capacity for manipulation of the droplets after the dispensing process using an electric field between the nozzle and the substrate. We expect our method to pave ways for new developments in drop-on-demand devices, for example, for dosing of reactive and temperature-sensitive solutions, emulsions and suspensions, for the release of droplets to be used as small reactors or for placing miniature samples onto plasmonically and/or electrochemically active micro- and nanostructure for their chemical analysis within microfluidic biotechnological devices.

## Methods

**Material.** Silicone oil (Silicone Oil AP 1,000, viscosity $\mu = 1,000$ mPa s, density $\rho = 1.09$ g cm$^{-3}$; Silicone Oil AP 100, $\mu = 100$ mPa s, $\rho = 1.06$ g cm$^{-3}$; Silicone Oil AR 20, $\mu = 20$ mPa s, $\rho = 1.01$ g cm$^{-3}$), methyltrichlorosilane, fluosilicic acid, glycol, glycerine and the surfactant Tween 60 were obtained from Sigma-Aldrich Co. and used as received. Methyltrichlorosilane and fluosilicic acid were used to form a hydrophobic layer on the glass capillaries. The interfacial tension between water and silicone oil was adjusted by Tween 60; the viscosity of the dispensed liquids was varied by addition of glycerine.

**Instruments and characterizations.** Hypodermic stainless needles (B. Braun Co.) of corresponding inner radius were polished to a flat end using sand papers to obtain nozzles with $R_o$ of 550, 175, 90 and 65 µm. The drainage rate of the dispensed liquid was controlled by a computer-controlled syringe pump with resolution of 83 nl (GeSIM XP3000, GESIM, Dresden, Germany). Pulled glass capillaries were used as nozzles with $R_o$ between 5 and 20 µm. To prevent the backward flow of the aqueous droplet along the outer surface of the hydrophilic glass pipette, the outer surface and the area surrounding the orifice was hydrophobized by a thin film of fluorinated silane formed by dipping the capillary into fluorinate silane collosol, which consist of methyltrichlorosilane, fluosilicic acid and deionized water in the molar ratio of 5:2:25. For these capillaries, the liquid flow was controlled by a syringe pump developed in-house with a resolution of 10 pl. It consists of a microlitre syringe for gas chromatography (10 µl, 700 Series, Hamilton Microliter Syringe, piston of radius 225 µm) connected to a piezoelectric motor (MS30 precision actuator and PS30 distance measurement system, CU30 controller, mechOnics AG, Munich, Germany) with minimum translation step of 50 nm. A high-speed camera (Neo sCOMS, Andor, EU) connected to an inverted optical microscope (ECLIPSE, TE2000U, Nikon, Japan) was used to record the shrinkage of the drops. $R_d$ and $V_d$ were obtained by image analysis and comparison to the known radius of the nozzles.

**Data availability.** The data that support the findings of this study are available from the corresponding author on request.

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

## Acknowledgements

Y.Z. thanks Alexander von Humboldt Foundation for the financial support. B.Z. thanks Sino-German (CSC-DAAD) postdoctoral scholarship for funding a research stay in Oldenburg. We thank Gerd Gertjegerdes, Saustin Dongmo, Julia Witt and Dr Carsten Dosche for helpful discussion and their help in building the experimental set-ups.

## Author contributions

Y.Z. and B.Z. equally contributed to the experimental work and data evaluation. Y.Z., B.Z. and G.W. conceived and designed the experiments; Y.Z., B.Z. and Y.L. designed and

prepared the experimental set-up; Y.Z. and G.W. wrote the paper. All authors analysed the data, discussed the results and commented on the manuscript.

## Additional information

**Competing financial interests:** The authors declare no competing financial interests.

