## [Peer Review File · Nature Communications]

Reviewers' comments:

Reviewer #2 (Remarks to the Author):

Second review of manuscript 15091754, "Hydrodynamic Dispensing and Electrical Manipulation of Ultrasmall (Attoliter) Droplets," submitted to Nature Nanotechnology by anonymous.

The revised manuscript is much improved compared to the original submission; I commend the authors for the additional work they have performed. There are still, however, unresolved gaps in the manuscript. I could recommend publication in Nature Nanotech provided the authors address the following points.

1) The new Fig 2 is crucial since it provides empirical evidence corroborating the authors' mechanistic interpretation of the underlying driving forces for the daughter droplet generation, but I suspect many readers will find it confusing (as I did). First, it is unclear from the figure and caption what is different from the top capillary number plot (2A) and bottom one (2C). The reader has to dig through the text to find out that actually several parameters were different, including the orifice diameter and viscosity. More importantly, it is completely unclear why the experimental data have to be separated into two different plots. The authors state "The existence of a critical capillary number is evident." Actually, they appear to observe two different critical capillary numbers: close to 2 for the top figure, and about 0.45 for the bottom figure. Why? What is different about the two sets of experiments that yield different critical capillary numbers? Why can't they plot all of the data on one large plot? The authors must clarify this point; as presented they certainly can't claim that they have identified "a" critical capillary number since there is more than one.

2) Likewise, the same sort of confusion pertains to Figs. 2B and 2D. Why are the data separated? What conditions were varied? Given the huge disparity in the ordinate scales, it appears that the data simply won't collapse in a way that would convince readers that the scaling prediction actually captures the droplet size.

3) Minor point: it is unclear what the x's mean in Fig 2. How does one measure the droplet size of an "attempted" droplet dispensing?

REVIEWERS' COMMENTS:

Reviewer #2 (Remarks to the Author):

Third set of comments on "Hydrodynamic Dispensing and Electrical Manipulation of Ultrasmall (Attoliter) Droplets" by anonymous.

I find that this third revision of the manuscript is much improved compared to the second revision. The confusion regarding the disparate critical Capillary numbers for different experimental geometries still leaves many unresolved questions, but I think that this paper will set the stage for more detailed investigation of this question. In regard to the overall picture, I should reiterate my initial surprise that draining the liquid quickly could initiate daughter droplet pinch-off, and I suspect that many other readers will be similarly surprised. I recommend publication in Nature Comm, although I recommend an extremely carefully editorial proofreading - there are still several grammatical awkward phrasings and outright spelling errors (e.g., 'fuction' in Fig. 4 caption).

Response to the reviewer comments

We would like to thank the reviewers for their time devoted to our manuscript. Below we provide a detailed answer to all of the comments (blue = reviewer comment, black our responses). The changes made to the manuscript are marked in yellow. Further changes in the structure of the manuscript were necessary as this manuscript was transferred from Nat. Nanotechnology to Nat. Communications. Those changes affect the arrangements of the material and as such are not marked separately to keep the focus on the main scientific issues.

The main concern with the reviewer was the existence of two critical capillary numbers, provoked by a misunderstandable phrasing in our manuscript. Each capillary setup has its own capillary number below which no droplets are formed. However, for one and the same setup, variations of drainage rate and viscosities etc. can be described by one number. We rephrased the corresponding sections and hope that it is clear to the reader now.

Reviewer #2 (Remarks to the Author):

Second review of manuscript 15091754, "Hydrodynamic Dispensing and Electrical Manipulation of Ultrasmall (Attoliter) Droplets," submitted to Nature Nanotechnology by anonymous.

The revised manuscript is much improved compared to the original submission; I commend the authors for the additional work they have performed. There are still, however, unresolved gaps in the manuscript. I could recommend publication in Nature Nanotech provided the authors address the following points.

Response: We thank the reviewer very much for the constructive and valuable comments which help us to improve our work.

1) The new Fig 2 is crucial since it provides empirical evidence corroborating the authors' mechanistic interpretation of the underlying driving forces for the daughter droplet generation, but I suspect many readers will find it confusing (as I did). First, it is unclear from the figure and caption what is different from the top capillary number plot (2A) and bottom one (2C). The reader has to dig through the text to find out that actually several parameters were different, including the orifice diameter and viscosity.

Response:

The former Figure 2 is now Figure 4 due to changes connected with the transfer to Nat. Commun.. To illustrate clearly the changes between the two setups we included two more panels with the drop geometries taken from video images (Figure 4a and 4d). While the mother drop is pinned at the outer diameter of the capillary in Figure 4a, it is pinned to the inner diameter in Figure 4d. We also added the detailed experimental settings for both data sets. We believe that the different experimental parameters are now very clear.

Changes made to the manuscript:

Fig. 4. Size of daughter droplets as a function of experimental parameters. **a**, Geometry when the mother droplet is pinned at the outer radius of the capillary. The shapes were extracted from video images; **b**, and **c**, experimental daughter drop sizes; **d**, geometry when the mother droplet is pinned at the inner radius of the capillary; **e**, and **f**, experimental daughter drop sizes. The size of the symbol in Fig. 4b and e is proportional to the size of the daughter droplets and the abscissa is given as dimensionless Ohnesorge number $Oh = \mu_2 [(\rho_1 + \rho_2)\sigma_{12}R_m]^{-1/2}$ comparing viscous, inertial and interfacial forces. Fig. 4c and f shows the linear relationship between R_d and $(\mu_2 Q / \sigma_{12})^{1/2}$. Crosses indicated dispensing under formation of several droplets of similar sizes (transition regime) where R_d is the average radius of the several droplets formed due to the break up of the thin liquid filaments.

More importantly, it is completely unclear why the experimental data have to be separated into two different plots. The authors state "The existence of a critical capillary number is evident." Actually, they appear to observe two different critical capillary numbers: close to 2 for the top figure, and about 0.45 for the bottom figure. Why? What is different about the two sets of experiments that yield different critical capillary numbers? Why can't they plot all of the data on one large plot? The authors must clarify this point; as presented they certainly can't claim that they have identified "a" critical capillary number since there is more than one.

Response: We wanted to express that for each geometry of the setup, there is a critical Capillary number Ca^* . It summarized for this setup the influence of drainage rate and viscosities and interfacial tension. The numerical value for Ca^* is different for different geometries (see Figure 4a and d).

Changes made to the manuscript:

The experimental results show that the pinch off regimes, as well as the size of the daughter droplets is controlled by the capillary number of the particular system (Fig. 4). Two sets of experiments with different orifice sizes and other parameters were carried out. In the first set, micro droplets were generated in five different silicon oils which have different viscosities (0.1, 0.25, 0.5, 0.76 and 1 Pa s), densities (varied from 1.01 to 1.09 g/mL) and interfacial tensions with the sessile drop (varied from 0.009 to 0.01 mN/m). Drainage rates varied from 4 to 70 $\mu\text{L/s}$. The mother drop is initially settled on the outer circle of the capillary (Fig. 4a). The three-phase contact line contracts during the liquid drainage process and finally anchored on the inner surface of the orifice after the drainage. In the second set of experiments, micro droplets were generated in one silicon oil ($\mu_2 = 1 \text{ Pa s}$, $\rho_2 = 1.09 \text{ g/mL}$, $\sigma_{12} = 0.01 \text{ Nm}$) with a stainless steel capillary ($R_o = 65 \mu\text{m}$). R_m varied from 140 to 200 μm and Q varied from 10 to 50 nL/s. The mother drop is initially settled on the inner circle of the capillary and is anchored there during the entire drainage process (Fig. 4d). The evolution of the drop shapes for the two sets of experiments at drainage rates of 7.1 $\mu\text{L/s}$ and 43 nL/s are shown in Fig. 4a and d, respectively. Within both sets, the size of the daughter droplets increases with Ca (Fig. 4b and e). There is a critical capillary number Ca^* in each geometry which separates the regimes of no droplet formation and droplet formation (red dashed lines in Fig. 4b and e). Due to the significant difference of the geometries, the value of Ca^* for the two sets are different.

2) Likewise, the same sort of confusion pertains to Figs. 2B and 2D. Why are the data separated? What conditions were varied? Given the huge disparity in the ordinate scales, it appears that the data simply won't collapse in a way that would convince readers that the scaling prediction actually captures the droplet size.

Response:

[The former Figure 2 is now Figure 4 due to changes connected with the transfer to Nat. Commun.]. This comment touches the same issue as comment 1). Each geometry has its critically Capillary number Ca^* that summarizes the effect of drainage rate and viscosities and interfacial tension for a particular geometry. Therefore, the results of two sets of experiments were shown in two plots. In fact the general concept works over a wide range of experimental conditions which necessitates different ordinate scales. Actually, we consider that as a strength of the new concept.

Changes made to the manuscript: see response to comment 1)

3) Minor point: it is unclear what the x's mean in Fig 2. How does one measure the droplet size of an "attempted" droplet dispensing?

Response: We have deleted the word "attempts" because it was misleading. The "x" refers to a regime where several droplets of similar sizes are formed.

Changes made to the manuscript:

Figure Caption 4 (was Figure 2 in the previous version)

Crosses indicated dispensing under formation of several droplets of similar sizes (transition regime) where R_d is the average radius of the several droplets formed due to the break up of the thin liquid filaments.

REVIEWERS' COMMENTS:

Reviewer #2 (Remarks to the Author):

Third set of comments on "Hydrodynamic Dispensing and Electrical Manipulation of Ultrasmall (Attoliter) Droplets" by anonymous.

I find that this third revision of the manuscript is much improved compared to the second revision. The confusion regarding the disparate critical Capillary numbers for different experimental geometries still leaves many unresolved questions, but I think that this paper will set the stage for more detailed investigation of this question. In regard to the overall picture, I should reiterate my initial surprise that draining the liquid quickly could initiate daughter droplet pinch-off, and I suspect that many other readers will be similarly surprised. I recommend publication in Nature Comm, although I recommend an extremely carefully editorial proofreading - there are still several grammatical awkward phrasings and outright spelling errors (e.g., 'fucntion' in Fig. 4 caption).

Response: we thank the reviewer very much for the successive constructive comments on our work, which helps us a lot to improve our work. We have carefully checked the language to meet the standard of Nature Communications.